

# A technical review and guide to RNA fluorescence in situ hybridization

Alexander P. Young[1], Daniel J. Jackson[2] and Russell C. Wyeth[1]

[1] Department of Biology, St. Francis Xavier University, Antigonish, NS, Canada
[2] Department of Geobiology, Georg-August Universität Göttingen, Göttingen, Germany

## ABSTRACT

RNA-fluorescence in situ hybridization (FISH) is a powerful tool to visualize target messenger RNA transcripts in cultured cells, tissue sections or whole-mount preparations. As the technique has been developed over time, an ever-increasing number of divergent protocols have been published. There is now a broad selection of options available to facilitate proper tissue preparation, hybridization, and post-hybridization background removal to achieve optimal results. Here we review the technical aspects of RNA-FISH, examining the most common methods associated with different sample types including cytological preparations and whole-mounts. We discuss the application of commonly used reagents for tissue preparation, hybridization, and post-hybridization washing and provide explanations of the functional roles for each reagent. We also discuss the available probe types and necessary controls to accurately visualize gene expression. Finally, we review the most recent advances in FISH technology that facilitate both highly multiplexed experiments and signal amplification for individual targets. Taken together, this information will guide the methods development process for investigators that seek to perform FISH in organisms that lack documented or optimized protocols.

## INTRODUCTION

Fluorescence in situ hybridization (FISH) is a powerful tool to visualize target DNA sequences or messenger RNA (mRNA) transcripts in cultured cells, tissue sections or whole-mount preparations. FISH functions via the principles of nucleic acid thermodynamics whereby two complementary strands of nucleic acids readily anneal to each other under the proper conditions to form a duplex (RNA:RNA or DNA:DNA), known as a hybrid (*Felsenfeld & Miles, 1967*). Under energetically favorable conditions, strands of RNA and DNA can also anneal to form DNA:RNA hybrids (*Rich, 1959*, *1960*; *Milman, Langridge & Chamberlin, 1967*). These phenomena have facilitated the development of techniques that use either DNA or RNA probes to bind to DNA or RNA targets within a biological sample, a method broadly known as in situ hybridization (ISH). The earliest ISH protocols relied on radioactive probes that were costly, required long exposure times, and were hazardous to human health (*Gall & Pardue, 1969*;

Corresponding author
Alexander P. Young, ayoung@stfx.ca

*Pardue & Gall, 1969*). Probes that relied on fluorophores instead of radioactive isotopes were later developed and could be directly detected with fluorescence microscopy. Methods that employed these probes became known as FISH (*Rudkin & Stollar, 1977*). As FISH can be used to target DNA, modern FISH protocols can label positions of genes on chromosomes, diagnose diseases and identify microorganisms (*Kempf, Trebesius & Autenrieth, 2000*; *Wiegant et al., 2000*; *Hicks & Tubbs, 2005*). However, FISH has also been developed to target RNA and thus visualize gene expression in situ, herein referred to as RNA-FISH (*Singer & Ward, 1982*). More recently, computational and imaging technology has further driven the development of RNA-FISH to allow for the visualization and semi-automated quantification of individual mRNA transcripts (*Femino et al., 1998*; *Levsky et al., 2002*; *Raj et al., 2006*, *2008*). The use of RNA-FISH to visualize individual mRNA molecules in this fashion is currently known as single-molecule FISH (smFISH; *Femino et al., 1998*). Ultimately, there are several derivations of the original ISH method that have diverged to localize either DNA or RNA molecules with one of many detection methods. In this review, we focus on RNA-FISH methods.

As the number of FISH-based methods has increased, the number of published reagents, probe types and detection methods have also expanded. This rise in options has increased the complexity faced by a researcher when developing a new FISH protocol or attempting to adapt an established protocol for use with a non-conventional sample type. Furthermore, published protocols rarely clarify which components are essential, and which are "traditional" elements inherited from previous iterations of a protocol. Thus, for a newcomer seeking to repurpose a published protocol, it is often unclear which steps of a protocol may be critical to its success or which steps could be removed for their own purposes. Here we review the technical aspects of RNA-FISH, including, but not limited to, smFISH. Based on a critical analysis of some leading published methods, we summarize the technique with respect to commonly used reagents for tissue preparation, hybridization, and post-hybridization washing and provide explanations of the functional roles for each reagent. The purpose of this review is to draw common FISH variants and their rationales together to equip users with the knowledge to develop novel applications of RNA-FISH for unexplored sample types. Thus, we present a broad survey of published RNA-FISH protocols to educate new users and streamline the methods development process for both experienced and new investigators. It is worth noting the substantial overlap between many published ISH and FISH protocols with respect to tissue preparation, hybridization, and post-hybridization. We have drawn information from a broad selection of protocols which could also benefit the development of non-fluorescent (also known as chromogenic or colorimetric) ISH protocols (excluding probe generation and detection).

## SURVEY METHODOLOGY

To compare differences in modern FISH methodologies (tissue preparation, hybridization and post-hybridization), the literature was broadly surveyed using PubMed and Google Scholar to search terms including "FISH", "fluorescent", "fluorescence" and "ISH". We also cross-referenced each article to identify further relevant resources from the published

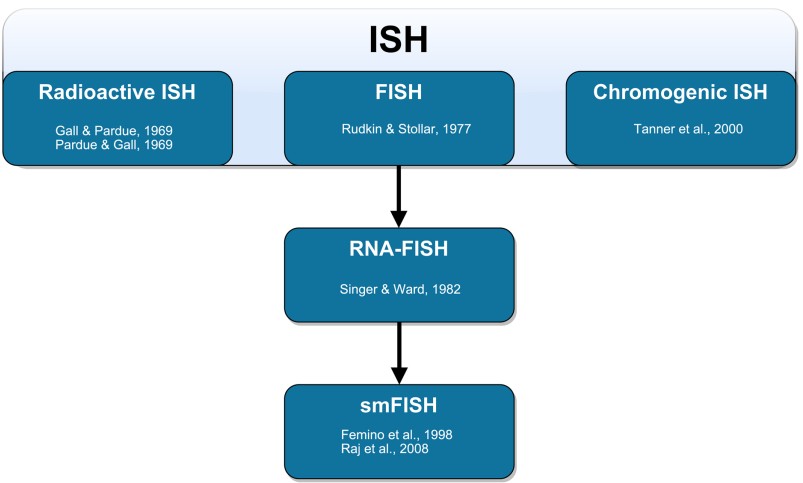

**Figure 1** Schematic representation of the technical development of fluorescent in situ hybridization (FISH). In situ hybridization (ISH) was first performed by *Gall & Pardue (1969)* using radioactive probes. Fluorescent ISH (FISH) against DNA was first performed by *Rudkin & Stollar (1977)*. FISH against RNA (RNA-FISH) was first performed by *Singer & Ward (1982)*. RNA-FISH that could be used to resolve individual mRNA transcripts was first performed by *Femino et al. (1998)* and later improved upon in whole mount tissue by *Raj et al. (2008)*. Horseradish peroxidase-based chromogenic (or colorimetric) ISH was later introduced by *Tanner et al. (2000)* as an alternative FISH without the need for a fluorescence microscope.            

literature. Manuscripts that included sufficiently detailed methods were selected for comparison. Generally, manuscripts from the last 10 years (after 2009) were preferred to reflect modern methods, however, we also include early works that heavily influenced the development of the technique. To support discussion of the commonly used reagents, we searched for manuscripts that specifically explained the mechanistic underpinnings of the reagents.

## The historical development of RNA-FISH

The method of labeling strands of nucleic acids in situ has undergone substantial development (Fig. 1). The earliest ISH techniques were documented in a pair of companion papers by *Gall & Pardue (1969)* and *Pardue & Gall (1969)*. *Gall & Pardue (1969)* used RNA-based probes to label DNA in oocytes of the toad *Xenopus*. *Pardue & Gall (1969)* used DNA-based probes to label DNA in the same cells from the same species. In both cases, these probes required autoradiography for visualization. The first fluorescence in situ detection of DNA with indirect immunofluorescence was performed by *Rudkin & Stollar (1977)* to label polytene chromosomes in *Drosophila melanogaster*. The authors used RNA probes with hapten-labeled nucleotides that could be targeted with rhodamine-labeled antibodies and subsequently visualized with a fluorescence microscope. These probes circumvented many of the disadvantages associated with autoradiography (*Bauman et al., 1980*; *Kislauskis et al., 1993*). Direct fluorescence in situ detection (of DNA) without the need for antibodies was later performed by *Bauman et al. (1980)*. The authors labeled mitochondrial DNA in the insect trypanosome *Crithilia*
*luciliae* using an RNA probe with rhodamine directly incorporated into the probe (RNA was oxidized with $NaIO_4$ and coupled to tetramethyl rhodamine thio-semicarbazide).

Although RNA-based probes had been used to this point, FISH had only been used to label DNA. *Singer & Ward (1982)* performed the first true RNA-FISH to visualize actin mRNA in a culture of chicken skeletal muscle. The authors used DNA probes labeled with biotin as a hapten (biotinylated dUTP was incorporated via nick-translation). Following hybridization, these probes were targeted with primary antibodies and then with secondary anti-biotin rhodamine-conjugated antibodies. The secondary antibody labeling allowed Singer and Ward to produce stronger fluorescence compared to the direct detection method of *Bauman et al. (1980)*. In this earlier development of RNA-FISH, probes had relied on either one fluorophore per probe molecule (and thus per hybridized transcript) or signal amplification using immunofluorescence. Neither of these methods produced adequately strong signals at a fixed fluorophore ratio per hybridized transcript that allows for absolute transcript quantification. Thus, only relative quantification of gene expression was possible.

Singer and colleagues later introduced the method of smFISH using multiple probes that were directly labeled with several Cy3 molecules per probe molecule. This method was sensitive enough to resolve individual mRNA transcripts (*Femino et al., 1998*). Due to the close proximity of fluorophores on the heavily labeled probe, the fluorophores underwent self-quenching (*Randolph & Waggoner, 1997*). This increased variability and interfered with quantification of the number of probe molecules bound to each transcript (*Femino et al., 1998*). In subsequent iterations of smFISH protocol development, the introduction of greater numbers of shorter singly-labeled probes resulted in labeling that was precise enough to allow for semi-automated quantification using image analysis software (*Raj et al., 2006*, *2008*; *Raj & Van Oudenaarden, 2009*; *Taniguchi et al., 2010*; *Lyubimova et al., 2013*). *Raj et al. (2006*, *2008)* used a series of 20-mer oligonucleotide probes to collectively span the length of the transcripts of interest. Each probe was tagged with a single Alexa 594 fluorophore at the 3′-terminus to yield a predictable number of fluorophores per transcript. *Raj et al. (2008)* found that this approach achieved a similar sensitivity in labeling individual transcripts compared to the method of *Femino et al. (1998)*, however, the newer method could more unambiguously discriminate between signal and background and had a simplified probe synthesis process. In parallel developments, other protocols were established using multiple nucleic acid-based probes with different fluorophores to measure the expression of multiple genes within individual cells (*Levsky et al., 2002*; *Raj & Van Oudenaarden, 2009*). smFISH has also been paired with immunofluorescence and flow cytometry to simultaneously measure mRNA and protein abundance (*Yoon, Pendergrass & Lee, 2016*; *Arrigucci et al., 2017*; *Eliscovich, Shenoy & Singer, 2017*).

## Technical aspects of FISH

Many permutations of the FISH methodology exist for a variety of niche purposes (*Volpi & Bridger, 2008*). Despite the range of techniques available, there is a core set of processing steps which are common to most: tissue preparation (pre-hybridization), hybridization

and washing (post-hybridization). These processes are essential to a FISH protocol, and each requires specific reagents to be effective. Generally, the required reagents are similar for cytological, histological and whole-mount preparations. However, there are some differences which are highlighted below. Note that the design and synthesis of a probe or multiple probes is also a critical phase of any ISH experiment that we will not discuss in depth here. However, characteristics such as the GC content, the propensity to form secondary structures, the overall length and specificity and probe quantity and quality must be considered (*Kucho et al., 2004*). It should be noted here that the use of purely synthetic oligonucleotide probes and short PCR-derived probes are gaining popularity over in vitro transcription-derived probes that span the majority of a transcript. Synthetic probes give the user great control over probe characteristics that affect hybridization (*Beliveau et al., 2012*, *2018*; *Bienko et al., 2013*) and omit the standard practice of cloning the target gene which delays the FISH process.

### Tissue preparation and permeabilization

Tissue preparation is one of the most critical aspects of a FISH protocol. Tissue preparation typically comprises both fixation and tissue permeabilization, and the balance of these is important in determining the degree of probe penetration as well as the morphological integrity of the sample. Prior to fixation, and critical for some species and sample types while less important for others, is the issue of relaxation of the sample of interest; a clear FISH signal can be obscured or rendered uninterpretable if it is concealed by a contracted morphology. Muscle relaxants are extremely species-specific and beyond the scope of this review, however an adequately relaxed tissue preparation (especially for whole-mounts) will make the visualization and interpretation of any signal significantly easier. We encourage the reader to survey the literature for appropriate relaxants for their species of interest. The most common fixatives are 4% formaldehyde or paraformaldehyde (PFA) in phosphate buffered saline (PBS; *Nakamura, Nakamura & Hamada, 2013*; *Neufeld et al., 2013*; *Kernohan & Bérubé, 2014*; *Shiura et al., 2014*; *Oka & Sato, 2015*; *Thiruketheeswaran, Kiehl & D'Haese, 2016*). Formaldehyde is a crosslinking fixative that forms covalent links between macromolecules such as lipids, peptides and DNA; this creates a mesh inside the cells or tissues to hold their components in place and minimize enzymatic degradation over time (*Eltoum et al., 2001*). PFA solutions produced from a powder will contain pure fixative, however, prepared 4% PFA solutions will produce polymers over time and become less effective as the polymers precipitate from the solution (*Thavarajah et al., 2012*). Thus, PFA solutions should be made fresh for each experiment. Alternatively, commercial formalin contains 37% monomeric formaldehyde in water and is supplemented with 10% methanol as a stabilizer to prevent polymer formation. Thus, a 1:10 dilution of commercial formalin solution is a common substitute for 4% PFA that does not require fresh preparation for each experiment (*Thavarajah et al., 2012*).

Fixation protocols are generally consistent among cytological, histological and whole-mount preparations, although whole mounts generally require longer treatments to ensure complete penetration of the fixative. Fixation protocols often consist of a treatment with 4% PFA or formaldehyde in PBS for varied lengths of time and temperatures

(Table S1). The following examples, and the link between sample size and density (larger and more dense samples need longer fixation) can provide some scope when estimating a fixation duration for other sample types. Optimal fixation of planarian worms is achieved with 4% formaldehyde for 20 min (*Pearson et al., 2009*; *Rink, Vu & Alvarado, 2011*). For bacterial species or eukaryotic cells, 4% PFA is used to fix cells for as little as 10 min or as much as 90 min (*Shaffer et al., 2013*; *Skinner et al., 2013*; *Chen et al., 2015*; *Wang et al., 2015*; *Aistleitner et al., 2018*; *Cardinale et al., 2018*; *Rocha, Almeida & Azevedo, 2018*). Fruit fly (*Drosophila melanogaster*) embryos are typically fixed in 4% PFA for 20–30 min (*Hauptmann et al., 2016*; *Jandura et al., 2017*; *Little & Gregor, 2018*; *Szabo et al., 2018*). Zebrafish (*Danio rerio*) embryos and the annelid *Platynereis dumerilii* can be suitably fixed in 4% PFA for 2 h at room temperature (*Jékely & Arendt, 2007*; *Steinmetz et al., 2011*) but can alternatively be fixed overnight at 4 °C (*Oxtoby & Jowett, 1993*; *Lauter, Söll & Hauptmann, 2011a*, *2011b*; *Marra et al., 2017*). Arms of the brittle star (*Amphiura filiformis*) are also sufficiently fixed in 4% PFA overnight at 4 °C. Embryos of the brachiopods *Terebratalia transversa* and *Novocrania anomala* should be fixed in 4% formaldehyde for 4 h (*Schiemann et al., 2017*; *Gąsiorowski & Hejnol, 2019*). The starlet sea anemone (*Nematostella vectensis*) and an acoelomorph worm (*Convolutriloba longifissura*) have been successfully fixed for ISH with 3.7% formaldehyde supplemented with 0.3% glutaraldehyde which is another strong cross-linking agent (*Finnerty et al., 2003*; *Martindale, Pang & Finnerty, 2004*; *Hejnol & Martindale, 2008*). Note, however, that glutaraldehyde is known to increase autofluorescence, at least with immunohistochemistry protocols. Whole mouse brains are often fixed in 4% PFA for up to 6 h at room temperature or overnight at 4 °C, although fixation of brain tissue is recommended not to exceed 24 h (*Kernohan & Bérubé, 2014*; *Kasai et al., 2016*; *Lanfranco et al., 2017*; *Hua et al., 2018*).

As an alternative to formaldehyde, some protocols employ alcohol-based fixation using either ethanol (*Schurter, LeBrun & Harrison, 2002*) or methanol (*Legendre et al., 2013*). Ethanol and methanol are coagulant fixatives that replace free water in the tissue to dehydrate cells and destabilize hydrophobic and hydrogen bonds (*Eltoum et al., 2001*). Alcohol-based fixation is common for cultured cells and ice-cold (−20 °C) ethanol and methanol have been used to fix multiple cultured cell lines in as little as 10 min (*Shaffer et al., 2013*). To fix tissue sections or whole-mounts, alcohol is commonly combined with other fixatives such as formaldehyde (*Finnerty et al., 2003*; *Martindale, Pang & Finnerty, 2004*; *Hejnol & Martindale, 2008*; *Pearson et al., 2009*). Although methanol has been used successfully with immunofluorescence (*Levitt & King, 1987*), methanol has a propensity to disrupt native protein structure and is generally not recommended for use in multiplex FISH and immunohistochemistry (*Fowler et al., 2011*). Methanol will strip membrane lipids to improve permeability (*Hoetelmans et al., 2001*) and ethanol can strip the external wax and lipids from plant tissues (*Bleckmann & Dresselhaus, 2016*). Thus, if cross-linking is also desired, formalin may improve tissue permeability over PFA due to the added methanol. For example, fixative solutions that contain alcohol and formaldehyde improve permeability in gram-positive bacterial preparations and may

retain higher DNA quality in cytological preparations (*Manz et al., 1994*; *Shaffer et al., 2013*).

Beyond the choice of fixative, fixation temperature can also have a substantial impact on the final tissue quality (*Fox et al., 1985*; *Thavarajah et al., 2012*). With the use of formaldehyde, heat can accelerate the fixation process; although heat also increases the release of formaldehyde fumes which are hazardous to human health (*Fox et al., 1985*; *Titford, 2001*). Additionally, heat can denature proteins and cause a loss of antigenicity which would negatively affect multiplex FISH and immunohistochemistry (*Fowler et al., 2011*). For nucleic acid visualization, reduced temperatures of 4 °C have been shown to preserve RNA throughout the fixation process (*Bussolati et al., 2011*). Additionally, ice-cold solutions of alcohol fixatives are recommended as the reduced temperatures will reduce the risk of over-permeabilization and subsequent leakage of target molecules.

Following fixation, samples are generally permeabilized to allow for proper penetration of hybridization reagents. Detergent treatment of fixed tissue is commonly employed at a concentration of 0.1% as it substantially improves permeability of the tissues via disruption of cellular membranes. The use of Tween-20 is common but other detergents including sodium dodecyl sulfate and Triton X-100 can also be used. The detergent 3-[(3-cholamidopropyl)dimethylammonio]-1-propanesulfonate (CHAPS) is commonly used as an additive detergent to multiplex FISH with immunofluorescence as it effectively protects the native structure of proteins (*Meyer, Garzia & Tuschl, 2017*; *Sepsi et al., 2018*). Whole-mount preparations generally require stronger detergent treatments compared to cytological preparations or sectioned tissue, thus, a more aggressive detergent treatment such as 4% Triton X-100 can be effective in whole-mounts (*Croll et al., 1999*).

Treatment with a nonspecific protease such as proteinase K will permeabilize the tissues after fixation and can also release target nucleic acid molecules from bound proteins (such as RNA binding proteins), making them more accessible for hybridization. There is generally an inverse relationship between fixation time and the strength of the proteinase treatment as more highly fixed tissues will require a stronger protease digestion to become permeable to the probe. A protease treatment is not always necessary for bacterial or eukaryotic cells as a detergent is usually sufficient, however, a brief treatment with a dilute solution of proteinase K should be considered if probe penetration is the suspected cause of an issue (*Carr et al., 2005*). Zebrafish embryos are treated with 10 μg/mL proteinase K in PBST for 2–20 min depending on the age (*Oxtoby & Jowett, 1993*; *Marra et al., 2017*). The same treatment is also recommended for snail embryos as well as whole-mount planarian worms and is sometimes applied to fruit fly embryos, although several other permeabilization strategies including acetone are also frequently used for *Drosophila* (*Paré et al., 2009*; *Pearson et al., 2009*; *Jackson, Herlitze & Hohagen, 2016*; *Hauptmann et al., 2016*; *Trcek et al., 2017*). Some protocols call for brain sections to be treated with proteinase K, however, many protocols omit this step as permeability is less of an issue with sectioned material (*Kasai et al., 2016*; *Hua et al., 2018*). The proteinase K treatment will require careful optimization as too little digestion will prevent probe penetration whereas too much digestion will destroy the morphology of the tissue and lead to increased background (*Tessmar-Raible et al., 2005*; *Bleckmann & Dresselhaus, 2016*).

As the degree of permeabilization with proteinase K can be a critical factor in the success of a FISH experiment, we recommend the use of accurately and consistently assayed batches of proteinase K enzyme such as supplied by New England Biolabs (Catalog: P8107S). As an alternative to proteinase K, pepsin has also been used to achieve more mild digestion of the tissue. Pepsin is preferred for cultured cells (*Buxbaum, Wu & Singer, 2014*) and tissue sections (*Moorman et al., 2001*; *Teng et al., 2017*) but potentially could be adapted to whole embryos. A treatment of 1 mg/mL pepsin in 0.01 N HCl is a common treatment, although the treatment length varies from 30 s to 10 min depending on the sample type (*Moorman et al., 2001*; *Buxbaum, Wu & Singer, 2014*; *Teng et al., 2017*).

Further permeabilization treatments are available as an alternative or an addition to protease treatments. A treatment of 1 M HCl at 37 °C for 30–50 min is effective to improve permeability of mycolic-acid-containing bacterial cells whereas other bacteria (including *Escherichia coli*) can be permeabilized in only 10 min (*Macnaughton, O'Donnell & Embley, 1994*). The addition of Triton X-100 or other detergent directly to the fixative in the initial fixation protocol has also been used to improve the permeability of bacterial cells through its interaction with cell envelope lipid molecules (*Jackson, Herlitze & Hohagen, 2016*; *Rocha, Almeida & Azevedo, 2018*). Protease-free detergent-based methods have also been successful for permeabilization of *Drosophila* embryos (*Boettiger & Levine, 2013*). Zebrafish embryos that are stored in methanol can be treated with 2% $H_2O_2$ for 20 min at room temperature to improve permeability (*Lauter, Söll & Hauptmann, 2011b*). This $H_2O_2$ treatment can also quench endogenous peroxidase activity and bleach tissues to reduce background in horseradish peroxidase-based assays (*Marra et al., 2017*). Organic solvents such as acetone have been used as an alternative to protease digestion of fragile embryos, and this method can also retain antigenicity for immunohistochemistry (*Nagaso et al., 2001*). In the preparation of whole-mounts with particularly tough integument, a digestion with 0.25% collagenase can be incorporated to improve permeability of dermal layers (*Wyeth & Croll, 2011*). Ultimately, careful optimization of the balance between fixation (strength, length and temperature thereof) and a proteinase based permeabilization is necessary to achieve a consistently high signal to noise ratio.

### Hybridization

For efficient and complete hybridization of probe to target, the optimal environment must be provided. The hybridization reaction can contain an array of different components (Table S1). In addition to the tissue, most documented hybridization solutions comprise a saline-sodium citrate buffer (SSC) with formamide, vanadyl-ribonucleoside complex (VRC), dextran sulfate, bovine serum albumin (BSA), competitor tRNA or DNA, and the probe (*Pinkel et al., 1988*; *Singer, 1998*; *Shaffer et al., 2013*; *Kernohan & Bérubé, 2014*; *Oka & Sato, 2015*). Alternative components include Denhardt's solution, ethylenediaminetetraacetic acid (EDTA), and Tween-20 (*Langenbacher et al., 2015*; *Parker et al., 2019*). In addition to the recipe of the hybridization solution, there are several reaction conditions that must be considered, including salt concentration, pH, and the temperature and duration of the hybridization reaction.

Formamide reduces the free energy of binding of nucleic acid strands to allow hybridization to take place at lower temperatures without a loss in specificity, thus improving structural preservation of the tissue (*McConaughy, Laird & McCarthy, 1969*; *Bauman et al., 1980*; *Blake & Delcourt, 1996*; *Fontenete et al., 2016*). As formamide stabilizes free bases and single-stranded DNA in solution, the melting temperature of DNA is decreased in a linear fashion by 2.4–2.9 °C per mole of formamide in the hybridization buffer (*Blake & Delcourt, 1996*). Formamide generally composes between 10% and 50% of the final volume of the hybridization buffer, but this range may be exceeded under specific circumstances (Table S1). Formamide is a toxic substance and, therefore, proper safety precautions must be made to avoid inhalation and direct contact with formamide (*Warheit et al., 1989*). Protocols that use safer alternatives to formamide, such as urea (*Sinigaglia et al., 2018*) have been developed but have yet to gain popularity (*Volpi, 2017*).

Vanadyl-ribonucleoside complex is an RNase inhibitor that is used to protect RNA-based probes or targets from enzymatic degradation (*Berger & Birkenmeier, 1979*; *Frazier & Champney, 2012*). VRC is typically added to the hybridization buffer at a final concentration of 10 mM as a precautionary measure. VRC is not compatible with solutions that contain EDTA as an equimolar concentration of a chelating agent will sequester the cations required for proper VRC function (*Puskas et al., 1982*). An RNase inhibitor is not absolutely necessary for successful ISH, but one should be considered if RNase contamination is a suspected problem.

Dextran sulfate is an anhydroglucose polymer that absorbs water molecules to reduce the free water in the reaction. This forces the probe and the target closer together, an effect referred to as molecular crowding, which enhances the rate of hybridization of the probe to the target (*Lederman, Kawasaki & Szabo, 1981*). Dextran sulfate can also improve fluorescent signals (*Van Gijlswijk et al., 1996*; *Franks et al., 1998*). Dextran sulfate is a synthetic analog of heparin which can also be used in the hybridization buffer and has also been reported to reduce background signal (*Singh & Jones, 1984*). Dextran sulfate is most often employed at a concentration of 50–100 mg/mL (Table S1; *Singer & Ward, 1982*; *Oka & Sato, 2015*; *Parker et al., 2019*).

Bovine serum albumin is used as a blocking agent to reduce background signal and thus improve the contrast of the probe (*Choo, 2008*). BSA blocks nonspecific binding of probe molecules to nucleic acid binding sites on proteins within the tissue as it can saturate the binding sites prior to the introduction of the probe. The use of BSA as a blocking agent may be especially important when using antibody-based detection methods. BSA is generally used at a concentration of 1 mg/mL (*Thiruketheeswaran, Kiehl & D'Haese, 2016*) up to 10 mg/mL (*Singer & Ward, 1982*).

Finally, sheared salmon sperm DNA or tRNA from *E. coli* or yeast is usually included in the hybridization buffer. The purpose of competitive nucleic acids is also to saturate nonspecific binding sites for probes to reduce background. Additionally, the competitor tRNA may protect target mRNA molecules via nonspecific blocking of RNase molecules that may have contaminated the solution. The optimal concentration of tRNA
within the hybridization buffer should be empirically determined as it may vary widely depending on the tissue sample and the probe (Table S1; *Langenbacher et al., 2015*; *Liu et al., 2019*).

There are several alternative hybridization buffer components that can be used to facilitate an optimal hybridization environment. Denhardt's solution is a broad blocking reagent composed of BSA, Ficoll type 400 and polyvinylpyrrolidone that can be used in place of BSA alone. EDTA is a chelating agent that can be added to a final concentration of 10 mM to remove free divalent ions such as magnesium. As EDTA can inactivate the VRC, these components are mutually exclusive.

When the reagent recipe has been established to create a supportive hybridization solution, the hybridization conditions must also be determined to facilitate optimal hybridization. We believe attention should be first given to the following parameters regarding hybridization: salt concentration, pH, hybridization temperature and duration of hybridization. Optimal hybridization will occur under conditions that allow the hybridization of the probe to the target but prevent the formation of nonspecific hybrids. Conditions that promote the sole formation of highly stable hybrids are known as highly stringent conditions whereas more permissive conditions that may allow the formation of nonspecific hybrids are considered less stringent. The stringency of the hybridization is affected by the concentration of salt in the hybridization solution (lower concentrations are more stringent) as well as the hybridization temperature (higher temperatures are more stringent). It is most common to keep the salt concentration constant (750 mM NaCl, 87.5 mM sodium citrate), with pH roughly between 7.0 and 8.5, and simply adjust the hybridization temperature to achieve the ideal stringency (*Pearson et al., 2009*; *Zhang et al., 2012*; *Jackson, Herlitze & Hohagen, 2016*). An initial denaturation step of 75 °C for 10 min can be used to denature all target and probe RNA to facilitate hybridization, the sample is then immediately adjusted to the designated hybridization temperature (*Jékely & Arendt, 2007*; *Jackson, Herlitze & Hohagen, 2016*). The optimal hybridization temperature is dependent on the length and composition of the probe. Although the hybridization temperature should be empirically optimized for every probe individually, short oligonucleotide probes (20–50 nucleotides) typically require lower hybridization temperatures of 37 °C whereas longer riboprobes of 1,000+ nucleotides may hybridize at temperatures >55 °C (*Pearson et al., 2009*; *Jackson, Herlitze & Hohagen, 2016*; *Fontenete et al., 2016*). Generally, the hybridization step cannot be over-incubated. Thus, an extended hybridization should be performed to allow probes to completely occupy available targets. Most often, 12–24 h is sufficient, regardless of the probe type (*Carleton et al., 2014*; *Jackson, Herlitze & Hohagen, 2016*; *Meyer, Garzia & Tuschl, 2017*; *Jandura et al., 2017*). Rapid hybridization has been achieved in cultured cells in as little as five minutes with the Turbo FISH method (*Shaffer et al., 2013*), but this is not a prudent point of entry for new protocols, especially for whole-mount material. Ultimately, salt concentration, hybridization temperature, and hybridization duration can be adjusted to create the optimal hybridization conditions with enough stringency to exclude non-specific labeling.

### Post-hybridization treatments

The purpose of the post-hybridization washes is to separate nonspecific hybrids and remove unbound probe molecules from the tissue to minimize background signal. Samples are typically subjected to increasingly stringent washes in SSC buffer containing formamide and a detergent (Table S1; *Jackson, Herlitze & Hohagen, 2016*; *Thiruketheeswaran, Kiehl & D'Haese, 2016*). Increased stringency can be achieved through sequential washes with incrementally reduced salt concentrations while the wash temperature is matched to the hybridization temperature (*Martindale, Pang & Finnerty, 2004*; *Hejnol & Martindale, 2008*; *Jackson, Herlitze & Hohagen, 2016*; *Schiemann et al., 2017*; *Gąsiorowski & Hejnol, 2019*). At the end of washing, the goal is to allow only the specific and stable hybrids to remain. A wash progression that finishes with a higher concentration of salt (or at a lower temperature, that is, lower stringency) will be less likely to denature and remove nonspecific hybrids, but also may preserve greater intensity of specific labeling.

In addition to nonspecific hybrids, autofluorescence and excessive background are issues that can diminish the visibility of true signal and influence the interpretation of the results. Treatment with 0.1% Sudan Black B in 70% ethanol is effective to minimize autofluorescence in sectioned brain tissue as well as cultured cells (*Oliveira et al., 2010*; *Qi et al., 2017*). If background signal is an issue, tissues can also be acetylated with 0.3% acetic anhydride in triethanolamine for 5–10 min (*Jackson, Herlitze & Hohagen, 2016*). This acetylation blocks positively charged proteins and amine groups (exposed during enzymatic permeabilization) in the tissue that could otherwise engage in electrostatic interactions with negatively charged probes.

The final process prior to visualization of results is tissue clearing to prevent lateral light scattering within the tissue (*Richardson & Lichtman, 2015*). Common methods of tissue clearing may involve either dehydration or hyperhydration of the tissue sample. An organic solvent-based method of clearing via a two-to-one mixture of benzyl benzoate and benzyl alcohol has been successfully used to visualize whole snail embryos (*Jackson, Herlitze & Hohagen, 2016*), however, the tissue must first be dehydrated with a graded series of ethanol. One potential issue with solvent-based clearing is that the dehydration process can cause substantial shrinkage of tissues (*Richardson & Lichtman, 2015*). Other methods of clearing that involve hyperhydration include the formamide-based ClearT (*Kuwajima et al., 2013*) as well as the urea-based CUBIC (*Susaki et al., 2015*; *Tainaka et al., 2014*). Methods of hyperhydration often involve large quantities of detergent and are most suitable when it is desirable to remove the majority of lipids from the tissue sample. A more advanced method of tissue clearing involves the use of anchor probes to fix the hybrids within a polymer matrix with subsequent digestion of non-RNA material (*Moffitt et al., 2016a*), however, this technique is most suitable for highly multiplexed FISH experiments.

## Probe selection and optimization for FISH

Probes are nucleic acid strands that may be composed of DNA, cDNA or RNA; they may be single-stranded or double-stranded and may vary in length from 20 bases to over

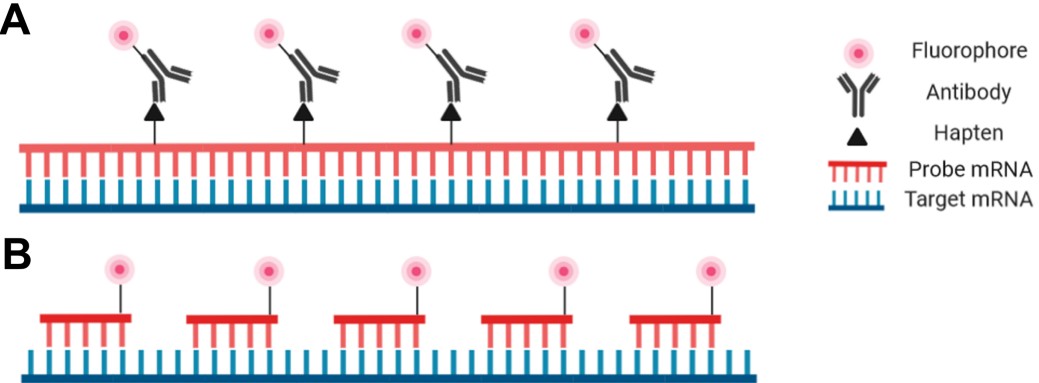

**Figure 2 Schematic representation of the riboprobe and oligonucleotide in situ hybridization probe types.** (A) Hapten-labeled RNA probes must be bound by an antibody labeled with a fluorophore to allow for visualization. (B) DNA oligomers directly labeled with a fluorophore can be directly visualized.

1,500 bases. Regardless of the probe type, the sequence of the probes must be complementary to the target sequence to ensure proper hybridization. Probes can be modified with a fluorophore directly attached to the probe to be detectable with fluorescence microscopy, or fluorophores may be covalently linked to an antibody that binds to an antigen incorporated into the probe (Fig. 2).

Despite the advantages and increasing popularity of chemically synthesized short probes (employed for example in smRNA-FISH), single-stranded RNA probes (riboprobes) of 500–1,500 bases are commonly employed as they are inexpensive and simple for a standardly equipped molecular biology laboratory to produce. Such riboprobes are typically generated through in vitro transcription of a target sequence that has been cloned. In this way target DNA sequences with flanking RNA polymerase promoters can be used with an appropriate RNA polymerase to produce single-stranded complementary RNA probes. Secondary detection is most common with riboprobes as nucleotides tagged with hapten molecules, such as digoxigenin, can be easily incorporated into the transcription reaction. The hapten molecules in the transcribed probe are then subsequently targeted by fluorophore-bound antibodies (Fig. 2). One advantage of riboprobes (rather than DNA-based probes) is that an RNase treatment can follow the post-hybridization step to reduce background. This is only appropriate with riboprobes as RNA:RNA hybrids are unaffected by RNases whereas DNA:RNA hybrids will be degraded (*Keller & Crouch, 1972*; *Donà & Houseley, 2014*). Note, however, that unintentional RNase contamination earlier in the protocol will be detrimental as single-stranded riboprobes are sensitive to RNases prior to hybridization.

The other prominent probe type in modern FISH protocols is the oligonucleotide probe—a cocktail of short single-stranded synthetic DNA probes that collectively span the length of the target (Fig. 2; *Femino et al., 1998*; *Raj et al., 2008*; *Zenklusen & Singer, 2010*). Each individual probe molecule can be labeled with a fluorophore on the 5′ end, the 3′ end, or both ends. A broad selection of fluorophores are available including Cy3, Cy5, Alexa fluor (Invitrogen, Carlsbad, CA, USA), and Quasar (LGC Biosearch Technologies,

Hoddesdon, UK) depending on the desired absorption/emission spectra, budget, or personal preference. Oligonucleotide probes can be advantageous for particularly challenging tissues as the small probes can penetrate the tissue more efficiently. Furthermore, as each oligonucleotide probe binds to the target, the transcript will relax and facilitate the hybridization of additional probe molecules (*Baker, 2012*). Oligonucleotides also have the highest specificity possible as they are less tolerant of mismatches that lead to nonspecific binding (*Hougaard, Hansen & Larsson, 1997*; *Insam, Franke-Whittle & Goberna, 2009*). One aspect of oligonucleotide probes that may deter new users is the level of difficulty associated with their production, or the high cost associated with outsourcing through a commercial supplier (*Raj et al., 2008*; *Zenklusen & Singer, 2010*).

## Controls for an ISH experiment

An often overlooked aspect of FISH experiments is how to employ controls to detect false positive results and to ensure that staining patterns represent genuine biological signal; if a staining pattern is observed following a FISH experiment, it may indicate successful hybridization, but it could also be the result of non-specific binding of the probe. Additionally, a lack of observable signal could mean that the mRNA target is not expressed, but it may also indicate a technical issue with the protocol despite the presence of the target. We would encourage creativity in carefully designing control experiments to identify the causes of undesired or absent results. Some potential control treatments are suggested below.

Several positive controls can potentially be used to verify both the efficacy of the FISH protocol and the expected behavior of all reagents. An example of a positive control to verify basic protocol function is to use a probe against a widely (temporally and in many tissue types) expressed gene such as actin or tubulin with a spatially discrete and predictable staining pattern (*Oschwald, Richter & Grunz, 1991*; *Kaplan et al., 1992*). It can also be informative to target specific genes that are only expressed in known tissue layers or cell types (e.g., neuronal- or epithelial-specific markers). Considering the relative ease and falling cost of generating transcriptome data, it is feasible to also select genes with high levels of expression from such data for use as positive controls. Finally, if no signal can be generated in situ with a positive control it may be informative to perform a simple in vitro dot blot. By spotting a diluted series of the probe onto a membrane and detecting these spots with the same reagents used in the in situ experiment any technical problems arising from the reagents can be ruled out or quickly identified.

Conversely, negative controls can identify nonspecific probe binding for direct labeling and nonspecific antibody binding for indirect labeling experiments. Parallel treatments in which one sample has been pre-treated with RNase will also indicate if the probe is binding exclusively to RNA (no signal is expected in the RNase treated sample). A similar treatment with DNAse will identify any binding to DNA. A sense probe can also be used in parallel with the normal antisense probe. A sense probe should not form a hybrid within the fixed tissue as it will not be complementary to a target, and thus can only produce non-specific binding. If sense and antisense probes are used in parallel and only

the antisense probe produces a signal, and all other controls are also verified, it is likely that the probe is specific and hybridized to the desired mRNA target (*Piette et al., 2008*). While this combination of controls is commonly employed in the literature and requested by reviewers, it has been reported that some genes are transcribed from both the sense and anti-sense DNA strands (*Katayama et al., 2005*; *Zhang et al., 2006*; *Hongay et al., 2006*; *Finocchiaro et al., 2007*). A combination of the above controls and experience with a range of probes against different genes will quickly give the user a sense of what is a general non-specific background versus a genuine biological signal.

## Recent advances in FISH protocol development

Since the inception of FISH, the core reagents required to perform the technique have remained relatively constant, however, significant advances have been made on the front of probe design and production, as well as signal amplification and detection (*Pichon et al., 2018*). Recent developments include improvements in the signal strength that can be achieved in small-scale experiments with complex whole-mounts (*Choi et al., 2016*, *2018*; *Marras, Bushkin & Tyagi, 2019*) as well as the high-throughput protocols that allow for visualization of thousands of transcripts in single cells with quantitative semi-automated data analysis (*Moffitt et al., 2016b*; *Eng et al., 2019*).

Amplification of FISH signal was first achieved through the use of fluorochrome-labeled tyramides that would accumulate at the site of the in situ hybrid due to the use of hapten-labeled probes and anti-hapten antibodies conjugated to horseradish peroxidase (*Raap et al., 1995*). This method of tyramide signal amplification for FISH is still frequently used to great effect in many sample types including whole-mount invertebrate embryos (*Martín-Durán et al., 2016*; *Schiemann et al., 2017*; *Gąsiorowski & Hejnol, 2019*) as well as vertebrate embryos and organs (*Lauter, Söll & Hauptmann, 2011a*, *2011b*; *Legendre et al., 2013*; *Row & Martin, 2017*). A more recent development for FISH signal amplification was introduced by *Choi et al. (2010)*, expanded on by *Marras, Bushkin & Tyagi (2019)* and is based on the hybridization chain reaction (HCR) introduced by *Dirks & Pierce (2004)*. In situ HCR uses RNA (*Choi et al., 2010*; *Choi, Beck & Pierce, 2014*) or DNA (*Dirks & Pierce, 2004*; *Choi et al., 2016*, *2018*) probes that carry overhang initiator sequences to initiate multiple chain reactions whereby multiple fluorophore-tagged DNA hairpins unfold and assemble into a chain in the vicinity of the probe. This effectively produces multiple strands of fluorophore-laden DNA that are tethered to the probe, thus substantially enhancing the signal. In situ HCR is a non-enzymatic method that boasts shorter protocol lengths (36 h) and does not exhibit the signal diffusion that has been associated with enzyme-based amplification and detection methods.

Methods for highly multiplexed FISH generally rely on either combinatorial (*Lubeck & Cai, 2012*; *Chen et al., 2015*; *Moffitt et al., 2016b*, *2018*) or sequential (*Lubeck et al., 2014*; *Shah et al., 2018*; *Eng et al., 2019*) labeling of individual transcripts using probes bearing different fluorophores to create RNA sequence-specific barcodes. Of the modern high-throughput multiplex approaches, multiplexed error robust FISH (MERFISH; *Chen et al., 2015*) and sequential FISH (seqFISH+; *Eng et al., 2019*) are two of the most

robust options. MERFISH utilizes multiple oligonucleotide probes per target, each probe with a 5′ and 3′ overhang readout sequence that can be separately targeted by a fluorophore-tagged secondary probe. SeqFISH+ also utilizes multiple singly-labeled oligonucleotide probes per transcript, however, the DNA:RNA hybrids are visualized, destroyed with DNase I, and then replaced using identical probes tagged with a spectrally distinct fluorophore to be imaged again. In both cases, the signals produced by all fluorophores are captured and the patterns are decoded using software to reveal the expression patterns of each gene. With these methods, 10,000 genes can be interrogated simultaneously within a single cell (*Eng et al., 2019*), or up to 40,000 cells within an 18 h measurement period (*Moffitt et al., 2016b*).

For most FISH protocols that involve labeling one or two target transcripts, qualitative analysis using confocal microscopy is sufficient, however, modern highly multiplexed FISH protocols require computer-assisted image analysis. Currently, single mRNA molecules can be detected using a standard epifluorescence microscope equipped with a charge-coupled device (CCD) camera, although data is typically collected from multiple optical slices using a confocal microscope (*Zenklusen & Singer, 2010*; *Skinner et al., 2013*). For analysis of standard smFISH experiments in cultured cells, it is generally possible to condense the full $z$-stack to a 2D image as for most genes, abundance is low enough that it is unlikely that two mRNA molecules will occupy the same position in the $x$–$y$ plane but differ in the $z$ plane (*Zenklusen & Singer, 2010*; *Trcek et al., 2012*). One of the most popular methods to extract data from these images involves fitting a 2D Gaussian mask over each diffraction limited spot to determine the exact signal intensity from each mRNA molecule (*Thompson, Larson & Webb, 2002*). Complex high-throughput datasets like those from MERFISH or seqFISH+ require specifically designed algorithms and substantial computational power to decode signals from hundreds of genes across multiple images from a single cell. The details of these analyses are beyond the scope of this review, but access to the computational pipelines is available through the respective MERFISH (*Moffitt et al., 2016b*) and seqFISH+ (*Eng et al., 2019*) publications.

Since the introduction of RNA-FISH, great progress has been made with respect to the number of targets that can be simultaneously visualized and quantified in situ. Substantial progress has also been made in terms of the complexity of tissues that can be processed, from cultured cells (*Singer & Ward, 1982*) to whole embryos (*Tautz & Pfeifle, 1989*). Whole mount FISH can be multiplexed to examine several transcripts simultaneously (*Meissner et al., 2019*) and MERFISH can be performed in tissue sections (*Moffitt et al., 2016a*). However, whole-mount techniques have not advanced to match what is possible in cultured cells. One requirement to close this gap is further development of imaging technology to visualize single transcripts using highly-multiplexed FISH in whole mounts. Furthermore, the development of signal enhancement methods such as branched DNA ISH (*Player et al., 2001*; *Battich, Stoeger & Pelkmans, 2015*) and HCR (*Choi et al., 2010*) will likely be a key to acquiring sensitive deep-tissue FISH signals in more complex samples.

## CONCLUSIONS

FISH is a powerful technique that can interrogate the spatial patterns and mechanisms of gene expression in biological systems on scales ranging from the single cell to tissue sections to whole organisms. When coupled with other modern methods that afford broad molecular insight (e.g., genomics, transcriptomics and gene editing), FISH can increase the precision of genetic information that can be ascertained from unconventional model organisms. However, establishing any kind of ISH method in an understudied system can be extremely time-consuming. This problem is compounded for the inexperienced user whose first step may be to consult an extremely varied, and at times contradictory, technical literature. In this review, we have attempted to summarize some of the main principles of FISH, and to emphasize those steps that are critical to success. As a starting method, we recommend 4% PFA or 3.7% formalin for fixation with 10 µg/mL proteinase K for permeabilization. The hybridization solution should contain at least formamide (generally 50%), dextran sulfate, and competitor DNA, but other ingredients and the duration of the hybridization are probe-dependent. Non-specific hybrids can then be removed during the post-hybridization washes using formamide and Tween-20 in SSC at the hybridization temperature, while progressively decreasing salt concentration. Finally, we have also highlighted some of the recent advances in the field and hope that in bringing these points to the attention of the reader, the process of FISH method development and optimization may be expedited.

### Funding

This work was supported by the Canadian Foundation for Innovation (Grant 19286 to Russell Wyeth), the Natural Sciences and Engineering Research Council of Canada (Discovery grant RGPIN-2015-04957 to Russell Wyeth), CGS-M and a Michael Smith Foreign Study Supplement (to Alexander Young), the Deutsche Forschungsgemeinschaft (Grant JA 2108/6-1 to Daniel Jackson), and St. Francis Xavier University. The funders had no role in study design, data collection and analysis, decision to publish, or preparation of the manuscript.

### Grant Disclosures

The following grant information was disclosed by the authors:
Canadian Foundation for Innovation: 19286.
Natural Sciences and Engineering Research Council of Canada: RGPIN-2015-04957.
CGS-M and Michael Smith Foreign Study Supplement.
Deutsche Forschungsgemeinschaft: JA 2108/6-1.
St. Francis Xavier University.

### Competing Interests

The authors declare that they have no competing interests.
## Author Contributions

- Alexander P. Young conceived and designed the experiments, performed the experiments, analyzed the data, prepared figures and/or tables, authored or reviewed drafts of the paper, and approved the final draft.
- Daniel J. Jackson analyzed the data, authored or reviewed drafts of the paper, and approved the final draft.
- Russell C. Wyeth conceived and designed the experiments, analyzed the data, authored or reviewed drafts of the paper, and approved the final draft.

## Data Availability

There is no data associated with this review article.

## Supplemental Information

Supplemental information for this article can be found online at http://dx.doi.org/10.7717/peerj.8806#supplemental-information.

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
