# Peer review of "A technical review and guide to RNA fluorescence in situ hybridization"

_PeerJ, doi:10.7717/peerj.8806_

## Round 0.1 · original submission · Major Revisions

The three expert reviewers and I agree that this review article will be a valuable contribution to the literature, in particular as it focuses on a neglected aspect of FISH experiments, how to design a protocol given the large existing literature. I concur, though, with all three reviewers that there are two additions to the manuscript that will be necessary before it is acceptable for publication: (1) a Conclusion section is needed; and (2) a discussion of recent advances to the FISH experiment (high-throughput variations, for example) should be added, to bring the review up-to-date and make the reader aware of the latest versions of the FISH experimental protocol. The reviewers provide substantial guidance to this recent literature and what should be added tot the manuscript. In addition, each reviewer provides several minor comments that will benefit the manuscript.

Thank you again for submitting this interesting and useful review article to PeerJ. I look forward to seeing your revised version.

Reviewer 1 ·

Basic reporting

Young and colleagues present a timely review that focuses on an important and largely unaddressed theme, a rational explanation of the diverse protocols for RNA FISH. The author’s description of diverse protocols with steps that have been inherited from earlier versions of uncertain utility to their current application, intermixed and often indistinct from elements critical to the success of the experiment, certainly resonated with my experience in this field in the past 15 years. Thus, I think the article will find a significant and interested audience. The author’s execution of this aim is largely successful in the manuscript.
The subject matter of RNA FISH taken alone is much to broad to easily cover comprehensively, even if the focus is just on protocols. Still, I find the treatment a little dated, as it misses a significant amount of innovation in RNA FISH protocols in the last 5 years which I think have made a dramatic impact on this field. I would recommend the authors to look into some of these works. I believe the push to higher throughput multiplexed detection and sensitive deep tissue detection has put greater pressure to optimize the protocols, and that these improvements to some degree benefit ALL forms of RNA FISH. Many of the modern protocols are variations of the earlier protocol varieties the authors already discuss, but I think their more stringent applications (and more recent use) can be used as a proxy to distinguish which protocols aspects are better. In particular, recent work from the Singer, Raj, Cai, Zhuang, Pelkmans, and Gregor labs.
some examples: Eng, …, Cai,. Nat. Methods 2017; Eng, Cai Nature 2019; Moffitt...Zhuang PNAS 2016a + 2016b; Shah…Cai Neuron 2016+2017; Sha…Cai Cell 2018; Battachi…Pelkmans Cell 2015; Chen…Zhuang 2015 Science, Battachi…Pelkmans Nat. Methods 2013; Eliscovich…Singer PNAS 2017; Moffitt, .... Zhuang, PNAS (2016a) + 2016b; Moffitt…, X. Zhuang Elife (2016), Shah…Cai, Neuron (2016) , (2017). Shah, … Cai, Cell. (2018); Chen…Zhuang Science 2015; Levesque Raj Nat. Methods 2015. Little…Gergor 2013-2015.

Minor comments by line:
Line 86 “…to label DNA the same cell types…” it seems a word is missing here
Line 111, 127: I would have liked to see Raj 2006 cited in the introduction on the smFISH. Moreso than the later Nature Methods paper (Raj 2008, cited line 111 and 127), this 2006 first introduced the significant tile and high contrast smFISH which expanded dramatically in the years following.
Line 156-8: A reference here could help, even if the author does not have space to elaborate more “…but we will not focus on this aspect here. However, characteristics such as the GC content, the propensity to form secondary structures, the overall length and specificity and probe quantity and quality must be considered”. The recent proliferation of purely synthetic probes in place of cloning derived probes has provided much greater control of characteristics such as GC content and the propensity to form secondary structures. E.g. Beliveau et al 2012 PNAS.
185 – I recommend boadening the time range, many modern protocols use shorter fixation for cultured cells, e.g. 10 minutes. (Chen…Zhuang 2015 ‘MERFISH’ on human cultured cells) or Skinner…Golding 2013 for e Coli.
~214 on alcohol fixation. Something could be said about “Turbo-FISH” protocol – a fast methanol based fixation.
229 – in discussion of ‘protection of native protein structure’ – it could be mentioned that alcohol solutions are generally denaturing and disrupt native protein structure, which can lead to loss of antigenicity for immuno FISH protocols (protein + RNA detection). The same is true of heat fix. GA fix can significantly reduce the permeability by providing to dense a crosslinking.
Line 244 – I would not recommend proteinase K treatment for Drosophila embryos. I think a more thorough search of the recent smFISH literature in Drosophila (not cited) will bear this out. See Pare…McGinnis 2009 Curr Bio, Boettiger…Levine 2009 Science, Boettiger…Levine 2013 Cell Reports, Little…Gregor Cell 2013, 2011 PLoS Bio, I could go on.
Some discussion of the Pepsin treatment (in place of proteinase K) as developed more recently by the Singer lab could go well alongside the proteinase K discussion. (Bauxbaum…Singer 2014 Science).
Line 286: 50% formamide – 10 -50% might be a more accurate way to put it. For example see the high resolution smFISH protocol described by Raj et al (2006 or 2008), and built on in recent smFISH work (see work from Long Cai and Xiaowei Zhuang’s groups on highly multiplexed smFISH protocols)
331 – have you validated the effect of EDTA on blocking RNases? RNase are not dependent on divalent cations the way DNases are.
364 – While I agree with the author that generally hybridizations can not go ‘too long’, it might be instructive to discuss the ‘5 minute’ ‘TURBO-FISH’ protocol as a nice example of rapid hybridization.
444 – recommendations for positive controls: Probes that are expected to give signal in every cell are not so easy to distinguish from background in every cell. I think it would be good to recommend in addition or in place of these controls, to use a patterned control in a heterogenous tissue (something say that should label epithielial but not mesenchyme, or neurons but not others) or a cell cycle probe in replicative tissue or replicative cultured cells KI67, or the histones loci make good probe targets as the mRNA is long or abundant but cell cycle specific in expression).
Some conclusion paragraph to pull the review together at the end would help the flow.

Experimental design

N/A (this is a review)

Validity of the findings

N/A this is a review -- see comments above for minor suggestions of accuracy.

Additional comments

Young and colleagues present a timely review that focuses on an important and largely unaddressed theme, a rational explanation of the diverse protocols for RNA FISH. The author’s description of diverse protocols with steps that have been inherited from earlier versions of uncertain utility to their current application, intermixed and often indistinct from elements critical to the success of the experiment, certainly resonated with my experience in this field in the past 15 years. Thus, I think the article will find a significant and interested audience. The author’s execution of this aim is largely successful in the manuscript.
The subject matter of RNA FISH taken alone is much to broad to easily cover comprehensively, even if the focus is just on protocols. Still, I find the treatment a little dated, as it misses a significant amount of innovation in RNA FISH protocols in the last 5 years which I think have made a dramatic impact on this field. I would recommend the authors to look into some of these works. I believe the push to higher throughput multiplexed detection and sensitive deep tissue detection has put greater pressure to optimize the protocols, and that these improvements to some degree benefit ALL forms of RNA FISH. Many of the modern protocols are variations of the earlier protocol varieties the authors already discuss, but I think their more stringent applications (and more recent use) can be used as a proxy to distinguish which protocols aspects are better. In particular, recent work from the Singer, Raj, Cai, Zhuang, Pelkmans, and Gregor labs.
some examples: Eng, …, Cai,. Nat. Methods 2017; Eng, Cai Nature 2019; Moffitt...Zhuang PNAS 2016a + 2016b; Shah…Cai Neuron 2016+2017; Sha…Cai Cell 2018; Battachi…Pelkmans Cell 2015; Chen…Zhuang 2015 Science, Battachi…Pelkmans Nat. Methods 2013; Eliscovich…Singer PNAS 2017; Moffitt, .... Zhuang, PNAS (2016a) + 2016b; Moffitt…, X. Zhuang Elife (2016), Shah…Cai, Neuron (2016) , (2017). Shah, … Cai, Cell. (2018); Chen…Zhuang Science 2015; Levesque Raj Nat. Methods 2015. Little…Gergor 2013-2015.

Minor comments by line:
Line 86 “…to label DNA the same cell types…” it seems a word is missing here
Line 111, 127: I would have liked to see Raj 2006 cited in the introduction on the smFISH. Moreso than the later Nature Methods paper (Raj 2008, cited line 111 and 127), this 2006 first introduced the significant tile and high contrast smFISH which expanded dramatically in the years following.
Line 156-8: A reference here could help, even if the author does not have space to elaborate more “…but we will not focus on this aspect here. However, characteristics such as the GC content, the propensity to form secondary structures, the overall length and specificity and probe quantity and quality must be considered”. The recent proliferation of purely synthetic probes in place of cloning derived probes has provided much greater control of characteristics such as GC content and the propensity to form secondary structures. E.g. Beliveau et al 2012 PNAS.
185 – I recommend boadening the time range, many modern protocols use shorter fixation for cultured cells, e.g. 10 minutes. (Chen…Zhuang 2015 ‘MERFISH’ on human cultured cells) or Skinner…Golding 2013 for e Coli.
~214 on alcohol fixation. Something could be said about “Turbo-FISH” protocol – a fast methanol based fixation.
229 – in discussion of ‘protection of native protein structure’ – it could be mentioned that alcohol solutions are generally denaturing and disrupt native protein structure, which can lead to loss of antigenicity for immuno FISH protocols (protein + RNA detection). The same is true of heat fix. GA fix can significantly reduce the permeability by providing to dense a crosslinking.
Line 244 – I would not recommend proteinase K treatment for Drosophila embryos. I think a more thorough search of the recent smFISH literature in Drosophila (not cited) will bear this out. See Pare…McGinnis 2009 Curr Bio, Boettiger…Levine 2009 Science, Boettiger…Levine 2013 Cell Reports, Little…Gregor Cell 2013, 2011 PLoS Bio, I could go on.
Some discussion of the Pepsin treatment (in place of proteinase K) as developed more recently by the Singer lab could go well alongside the proteinase K discussion. (Bauxbaum…Singer 2014 Science).
Line 286: 50% formamide – 10 -50% might be a more accurate way to put it. For example see the high resolution smFISH protocol described by Raj et al (2006 or 2008), and built on in recent smFISH work (see work from Long Cai and Xiaowei Zhuang’s groups on highly multiplexed smFISH protocols)
331 – have you validated the effect of EDTA on blocking RNases? RNase are not dependent on divalent cations the way DNases are.
364 – While I agree with the author that generally hybridizations can not go ‘too long’, it might be instructive to discuss the ‘5 minute’ ‘TURBO-FISH’ protocol as a nice example of rapid hybridization.
444 – recommendations for positive controls: Probes that are expected to give signal in every cell are not so easy to distinguish from background in every cell. I think it would be good to recommend in addition or in place of these controls, to use a patterned control in a heterogenous tissue (something say that should label epithielial but not mesenchyme, or neurons but not others) or a cell cycle probe in replicative tissue or replicative cultured cells KI67, or the histones loci make good probe targets as the mRNA is long or abundant but cell cycle specific in expression).
Some conclusion paragraph to pull the review together at the end would help the flow.

Reviewer 2 ·

Basic reporting

(1) The manuscript fails to include the recent development of the RNA-FISH technique. As a promising technique for RNA labeling and imaging, RNA-FISH has been improved in different ways to increase the throughput (e.g., Eng et al., Nature, 2019; Shah et al., Cell, 2018) and signal level (e.g., Marras et al., PNAS, 2019; Choi et al., Development, 2018). These new versions of RNA-FISH may point to the future direction of the field. Without mentioning these recent advances and discussing their technical aspects, the manuscript looks a bit outdated. Hence, the authors should include these new versions of RNA-FISH technique in their paper.

(2) Figure caption is missing for Figure 1. Although Figure 1 is a simple figure, it has only been cited once in the main text (line 83) without a detailed description. A caption is therefore required to help explain the whole figure. For example, the term “Chromogenic ISH” only appears in Figure 1, but not in the main text. With a figure caption, it is hard to understand what the term means.

(3) In lines 83 and 415, “(Figure 1)” and “(Figure 2)” should be “(Fig. 1)” and “(Fig. 2)”, respectively, according to the Instructions for Authors (https://peerj.com/about/author-instructions/#figures).

(4) In Tables 1 and 3, the temperature of each step is not listed.

Experimental design

no comment

Validity of the findings

The manuscript lacks a summary or conclusion part. A review article should not merely be a collection of previously published papers. As required by PeerJ, a Literature Review should have a conclusion part identifying unresolved questions/gaps/future directions about the field (https://peerj.com/about/author-instructions/#literature-review-sections). In this part, the authors may summarize their review of the technical aspects of RNA FISH and provide their view on the future development of the FISH technique. This will help the readers quickly understand the main points of the paper.

Additional comments

This paper reviews the technical aspects of RNA-fluorescence in situ hybridization (FISH), which serves as a powerful tool for visualizing the spatial distribution of specific RNA species in cultured cells, tissue sections or whole mount preparations. Focusing on the core steps of RNA-FISH procedure, i.e., tissue preparation, hybridization, and post-hybridization washing, the authors summarize reagents and experimental conditions commonly used in these steps and explain their functional roles. Also, the authors discuss probe selection and necessary controls for a robust FISH experiment.

In recent years, RNA imaging techniques, including RNA FISH, have been reviewed by multiple papers (Tutucci et al., Annu. Rev. Biophys., 2018; Pichon et al., Mol. Cell, 2018), most of which focused on comparing different RNA labeling and imaging techniques. However, a detailed review of the technical aspects of RNA FISH (or other RNA imaging techniques), which are critical for the success of an experiment, is still lacking. Hence, the information provided by this paper may help a wide range of investigators using RNA-FISH in their research. I think the manuscript is suitable for PeerJ. There are, though, some points that I would like the authors to address before publication:

(1) The manuscript lacks a summary or conclusion part. A review article should not merely be a collection of previously published papers. As required by PeerJ, a Literature Review should have a conclusion part identifying unresolved questions/gaps/future directions about the field (https://peerj.com/about/author-instructions/#literature-review-sections). In this part, the authors may summarize their review of the technical aspects of RNA FISH and provide their view on the future development of the FISH technique. This will help the readers quickly understand the main points of the paper.

(2) The manuscript fails to include the recent development of the RNA-FISH technique. As a promising technique for RNA labeling and imaging, RNA-FISH has been improved in different ways to increase the throughput (e.g., Eng et al., Nature, 2019; Shah et al., Cell, 2018) and signal level (e.g., Marras et al., PNAS, 2019; Choi et al., Development, 2018). These new versions of RNA-FISH may point to the future direction of the field. Without mentioning these recent advances and discussing their technical aspects, the manuscript looks a bit outdated. Hence, the authors should include these new versions of RNA-FISH technique in their paper.

(3) In some places of the paper, the same content has been reviewed twice, which is not necessary. For example, in “The development of RNA-FISH and smFISH” part, the first paragraph reviews the development of different RNA-FISH methods, while the second paragraph more reviews explicitly the development of different FISH probes. However, since the main difference between various RNA-FISH methods is their probes, the contents of the two paragraphs are highly overlapping. The authors should combine the two paragraphs and review the history of different RNA-FISH methods based on the development of different types of probes.

(4) In the last paragraph of the “Hybridization” part, where the authors discuss the hybridization conditions, the statement “three parameters regarding hybridization: salt concentration; hybridization temperature; and duration of hybridization” is not complete, as the pH value of the hybridization buffer is also an important parameter to affect the stringency of the hybridization (Choi et al., Development, 2016; Mannheim, Biochemicals Catalog, 1992). The authors should discuss how pH value affects hybridization in their paper.

Minor points:

(5) Figure caption is missing for Figure 1. Although Figure 1 is a simple figure, it has only been cited once in the main text (line 83) without a detailed description. A caption is therefore required to help explain the whole figure. For example, the term “Chromogenic ISH” only appears in Figure 1, but not in the main text. With a figure caption, it is hard to understand what the term means.

(6) In lines 83 and 415, “(Figure 1)” and “(Figure 2)” should be “(Fig. 1)” and “(Fig. 2)”, respectively, according to the Instructions for Authors (https://peerj.com/about/author-instructions/#figures).

(7) In Tables 1 and 3, the temperature of each step is not listed.

(8) In line 81, the subtitle “the development of RNA-FISH and smFISH” is not correct, as smFISH is a specific type of RNA-FISH. The authors may modify the subtitle to “the development of RNA-FISH”.

(9) In the last row of table 3, except for the first column, the duration of the final wash is missing.

(10) There are some language issues in the manuscript. Here are a few examples. The authors will have to make sure all such errors are corrected.

• In line 293, “…is used protect RNA-based probes” should be “…is used to protect RNA-based probes”.
• On the last page, “Table 3. Table 2. Representative panel of fluorescence…” should be “Table 3. Representative panel of fluorescence …”.


References:
Tutucci, Evelina, et al. "Imaging mRNA in vivo, from birth to death." Annual review of biophysics 47 (2018): 85-106.Salvatore A. E. Marrasa,b, Yuri Bushkina,c, and Sanjay Tyagi (2019).

Pichon, Xavier, et al. "A growing toolbox to image gene expression in single cells: sensitive approaches for demanding challenges." Molecular cell 71.3 (2018): 468-480.

Eng, Chee-Huat Linus, et al. "Transcriptome-scale super-resolved imaging in tissues by RNA seqFISH+." Nature 568.7751 (2019): 235.

Shah, Sheel, et al. "Dynamics and spatial genomics of the nascent transcriptome by intron seqFISH." Cell 174.2 (2018): 363-376.

Marras, Salvatore AE, Yuri Bushkin, and Sanjay Tyagi. "High-fidelity amplified FISH for the detection and allelic discrimination of single mRNA molecules." Proceedings of the National Academy of Sciences (2019): 201814463.

Choi, Harry MT, et al. "Mapping a multiplexed zoo of mRNA expression." Development 143.19 (2016): 3632-3637.

Mannheim, Boehringer. "Nucleic Acid Hybridization—General Aspects." Biochemicals Catalog (1992): 14-17.

Reviewer 3 ·

Basic reporting

See below.

Experimental design

See below.

Validity of the findings

See below.

Additional comments

In this Review, Young et al. present a brief historical account on the development of the most used RNA-FISH techniques, followed by a description of the most typical protocol steps, with emphasis on the rationale behind the use of the most commonly used reagents and procedures.

A review like this could be of use as an introduction for researchers that are interested in using RNA FISH but do not have prior experience in method development. In that sense, I think this article is worthwhile publishing. Below I list a series of comments for the authors to consider before publishing this piece in PeerJ:

1. Lines 110-111: “Since these seminal studies, the applications of smFISH have expanded to visualize differential gene expression within heterogeneous cells”. I think that using the term “heterogeneous cells” makes this sentence ambiguous. I think that the authors could change it to “(…) visualize gene expression heterogeneity at the single-cell level”.

2. Lines 112-113: The authors cite Shalek et al. 2014 when they talk about RNA-FISH multiplexing. But that reference uses mainly single-cell sequencing, using RNA-FISH only as a secondary confirmatory technique, I, therefore, think that this reference can be omitted.

3. Following comment #2, the authors mention smFISH multiplexing but they do not cite the most up-to-date RNA-FISH multiplexed techniques, such as MERFISH (Chen et al., Science, 2015, Moffitt et al., Science, 2018) or d seqFISH (Lubeck et al. Nat. meth., 2014; Eng et al, Nature, 2019). I think it is fitting to cite such techniques as the final evolution of RNA-FISH.

4. Lines 122-123: “Both methods had issues producing signal to noise ratios that could allow reliable quantification of transcript expression”. This phrase is vague. What “issues” is the author referring to? This could be solved by stating explicitly the problem of using a single fluorophore per mRNA (or a few, in case of antibodies): that probe binding to mRNA cannot be distinguished from non-specific binding to other cellular structures. smFISH allows obtaining the absolute number of molecules, while previous methods only could measure relative expression levels.

5. Lines 125-126: The authors mention self-quenching. Where do they get that reference? I know that Raj et al., Nat. meth., 2008 mention self-quenching as a drawback of the method used by Femino et al. and they cite Randolph et al., Nucleic Acids Res., 1997. The authors should cite either of those sources (or any source they used to back up their claim). Also, the authors may want to mention that probably the most important reason this method is not widely adopted is the difficulty to create such probes (As mentioned in Raj et al., Nat. meth., 2008). Self-quenching by itself does not avoid the method from reaching single-molecule resolution (as evidenced by the many works from the Singer lab)

6. Lines 130-132: The authors found that this approach was more effective in labeling individual mRNA targets compared to traditional probes that spanned the full length of the transcript”. Here again, the language is vague. What do the authors mean by “more effective”? Also, which comparison the authors mean? I do not remember Raj et al. 2008 making such a comparison. Also, please provide a reference when making claims.

7. Lines 135-137: The authors mention Stellaris as a product that was created based on smFISH technology. I find that only mentioning a single product seems to be unnecessary favoritism. I imagine there are other commercial products that are used for other types of RNA-FISH. If this is the case, It would be better the authors add a table for a series of technologies or remove the reference to commercial products altogether.

8. Line 270: ”(…) is time well spent”. It seems too colloquial and vague. It would be better to say explicitly what can be improved, for example, “it is necessary to obtain a high signal to noise ratio” or something like that, depending on what the authors want to emphasize.

9. Line 284: “(…) reduces the annealing temperatures of nucleic acid strains”. That’s confusing. The annealing temperature is an experimental parameter that can be varied. A more appropriate parameter to mention would be the free energy of binding (which depends on the temperature).

10. Lines 343-345: It is true that the salt concentration is generally kept constant. Temperature is changed often, but also formamide concentration is often changed (see multiple reviews from Raj lab.)

11. Line 426: I would suggest the authors to add references to works where the authors use the comparison of the foci intensity in samples with or without the transcript of interest (for example, Skinner et al. Nat. Protoc., 2013) or the presence of a single foci intensity peak (Trcek et al., Nat. Protoc., 2012)

12. Finally, the review would benefit from a “Conclusion” or “Discussion” section. Also, it would also benefit from a section describing, even briefly, the type of analysis researchers perform with the data (particularly for smFISH).

---

## Round 0.2 · accepted · Accept

I appreciate the careful attention that you have paid to the reviewers' comments.

Reviewer 1 ·

Basic reporting

The authors have addressed my concerns.

Experimental design

The authors have addressed my concerns.

Validity of the findings

The authors have addressed my concerns.

Additional comments

The authors have addressed my concerns.